# Precision Medicine Revolutionizing Esophageal Cancer Treatment: Surmounting Hurdles and Enhancing Therapeutic Efficacy through Targeted Drug Therapies

Poojarani Panda [1], Henu Kumar Verma [2] and Lakkakula V. K. S. Bhaskar [1,*]

1   Department of Zoology, Guru Ghasidas Vishwavidyalaya, Bilaspur 495009, India; poojaranipanda28@gmail.com
2   Department of Immunopathology, Institute of lungs Health and Immunity, Comprehensive Pneumology Center, Helmholtz Zentrum, 85764 Munich, Germany; henu.verma@yahoo.com
*   Correspondence: lvksbhaskar@gmail.com

**Simple Summary:** Esophageal cancer is a challenging cancer with a poor prognosis. Targeted pharmacological therapy has emerged as a viable option for improving patient outcomes. This article examines the usage of numerous medications that target certain cancer growth pathways such as EGFR, HER-2, VEGFR, mTOR, and cMET. It also discusses the influence of medication resistance on therapy efficacy. While these targeted medicines do not promise a cure, they have the potential to improve survival rates in patients with esophageal cancer.

**Abstract:** Esophageal cancer is a formidable challenge in the realm of cancer treatment. Conventional methods such as surgery, chemotherapy, and immunotherapy have demonstrated limited success rates in managing this disease. In response, targeted drug therapies have emerged as a promising strategy to improve outcomes for patients. These therapies aim to disrupt specific pathways involved in the growth and development of esophageal cancer cells. This review explores various drugs used to target specific pathways, including cetuximab and monoclonal antibodies (gefitinib) that target the epidermal growth factor receptor (EGFR), trastuzumab that targets human epidermal growth factor receptor 2 (HER-2), drugs targeting the vascular endothelial growth factor receptor (VEGFR), mTOR inhibitors, and cMET inhibitors. Additionally, the article discusses the impact of drug resistance on the effectiveness of these therapies, highlighting factors such as cancer stem cells, cancer-associated fibroblasts, immune-inflammatory cells, cytokines, hypoxia, and growth factors. While drug targeting approaches do not provide a complete cure for esophageal cancer due to drug resistance and associated side effects, they offer potential for improving patient survival rates.

**Keywords:** esophageal cancer; targeted drug therapies; pathway targeting; drug resistance; patient survival rates



## 1. Introduction

Esophageal carcinoma (EC) is a malignant disease with devastating effects. It is the ninth most common cancer and ranks as the sixth leading cause of cancer-related deaths worldwide [1]. EC, which includes esophageal squamous cell carcinoma (ESCC) and esophageal adenocarcinoma (EAC), is a kind of cancer that develops in the esophagus [2]. If it occurs in the middle or upper portion, EC can be fatal [3]. The male-to-female ratio is 3:1 for ESCC, while for EAC, it is 6:1. However, this ratio varies significantly across different geographical locations [4,5]. Cohort studies indicate that the incidence of esophageal cancer increases with age, with the typical age of onset being 65 to 70 years [6]. EC is characterized by progressive dysphagia and weight loss. Dysphagia often presents with the vomiting of undigested food [7].

Tobacco smoking and excessive alcohol consumption, particularly when combined, are the primary risk factors for esophageal squamous cell cancer. EC is associated with gastroesophageal reflux disease and obesity [8]. A study suggests that squamous cell carcinoma is linked to an overrepresentation of C to A substitutions, which is more common among cigarette users [9]. Esophageal adenocarcinoma is associated with abnormalities in homologous recombination repair, a high mutational burden, or an aging-related C to A or T mutation pattern [10]. According to the World Health Organization, based on the Global Burden of Disease Study 2015, the estimated mortality rate from EC is 439,025 [11]. In 2018, there were approximately 500,000 cases of EC worldwide. Unfortunately, most patients are diagnosed at an advanced stage, resulting in a low overall survival rate of 20% at the end of five years [9]. However, multimodal therapy may lead to better outcomes, as it has shown some improvement in overall survival [12]. Patients diagnosed with ESCC or EAC often present at an advanced stage, leaving limited treatment options. Chemotherapy is employed as a palliative approach for end-of-life cancer patients [13]. Unfortunately, there has been little significant clinical advancement in EC treatment over the past three decades, resulting in modest improvements in survival rates. Recently, Ye et al. discovered that the median survival period of the EC-SPM group was 74.67 months in 73,456 patients with EC. Survivors of esophageal cancer who develop a second primary malignant cancer have a better prognosis, but they must undergo more rigorous treatment [14].

Chemotherapy has been a cornerstone in EC treatment, while molecular targeted therapy and immune checkpoint inhibitors have been investigated in preclinical and clinical trials [7,15,16]. Targeted molecular therapy serves as a vital adjunct to chemotherapy, although only a few targeted therapies are currently available in clinical practice [17]. Despite the initial success of targeted therapy in early-stage EC treatment, patients inevitably develop drug resistance during the course of treatment [18]. In addition to treatment resistance, recurrence and metastasis are the primary causes of treatment failure [19]. Precision medicine is used to diagnose cancer, design a patient's course of therapy, assess the effectiveness of that treatment, and determine the patient's prognosis. In early trials of personalized and targeted therapy, medication selection based on genetic data has produced positive outcomes. Most clinically authorized targeted medicines are now focused on targeting kinases [20].

Several monoclonal antibodies and tyrosine kinase inhibitors (TKIs) have been discovered to date. These drugs can be used alone or in combination with conventional treatments to improve the prognosis of ESCC patients. This review primarily focuses on the utilization of different drug types to target signaling pathways (EGFR, HER2, VEGF, HGF/cMET, mTOR) associated with EC and inhibit their activity. Additionally, we address challenges related to targeting, such as cancer heterogeneity and the molecular mechanisms underlying drug resistance. Furthermore, we discuss the side effects of drugs used in targeting and present future prospects.

## 2. Drugs Targeting the Key Signaling Pathway

ESCC is believed to result from the dysregulation of cell signaling networks rather than individual mutations, as supported by recent research. There is an increasing consensus that mutations primarily affect signaling pathways rather than specific genes [20]. Therefore, targeting drugs towards these specific signaling pathways is crucial to reducing the risk of cancer. Numerous monoclonal antibodies and tyrosine kinase inhibitors have been discovered to date, which can be utilized alone or in combination with conventional treatments to improve the prognosis of esophageal cancer [3] (Figure 1).

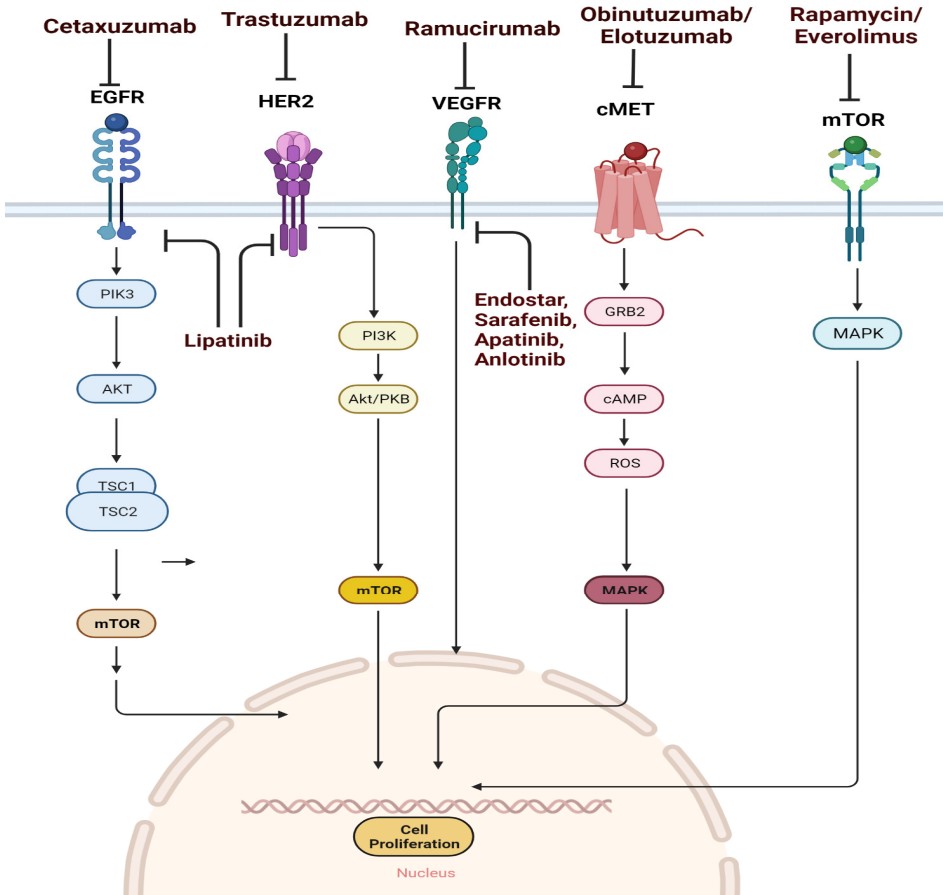

**Figure 1.** Schematic representation of medications targets the pathway and prevents downstream signaling.

Cetuximab and nimotuzumab are monoclonal antibodies that target the EGFR, while gefitinib and lapatinib are EGFR receptor tyrosine kinase inhibitors that bind to the RTK domain of EGFR, thereby downregulating signaling. Trastuzumab specifically targets HER-2 and increases the survival of patients. Ramucirumab is a monoclonal antibody that binds to VEGFR, while endostar and apatinib inhibit downstream signaling pathways. Rilotumumab inhibits the interaction of hepatocyte growth factor (HGF) with c-MET, and obinutuzumab blocks c-MET from binding to HGF. Rapamycin and everolimus inhibit mTOR signaling. All these drugs downregulate signaling and block esophageal cancer cell proliferation and division.

### 3. Drugs Targeting the EGFR Pathway

EGFR is a transmembrane growth factor receptor family that initiates signal transduction by activating a receptor-associated tyrosine kinase (RTKs) [21]. EGF binding to EGFR causes this receptor to homodimerize or heterodimerize with other ERBB members, namely HER2, which in turn activate the RAS-RAF-MEK-ERK-MAPK and PI3K-AKT-mTOR pathways, relaying downstream signals. Overexpression of EGFR contributes to carcinogenic effects [22,23]. Recently, several drugs with similar mechanisms have been discovered to block EGFR signaling, enabling the successful treatment of EC [24]. Few targeted drugs have been authorized by the FDA for the treatment of esophageal cancer, despite the fact that targeted therapy has emerged as a significant therapeutic option for many malignant tumors. Trastuzumab deruxtecan (HER2-targeting) has been authorized as a first-line therapy in conjunction with chemotherapy in patients with gastroesophageal adenocarcinoma and gastric cancer who test positive for HER-2 at this time. Ramucirumab (VEGFR-targeting) is the only second-line treatment for EAC that has received this ap-

proval [3]. Numerous monoclonal antibodies targeting EGFR have been designed, and inhibitors have been developed to inhibit EGFR signaling [25].

## 4. Anti-EGFR Monoclonal Antibodies

Anti-EGFR monoclonal antibodies have been widely explored and used in the treatment of different malignancies, including EC. According to reports, EGFR monoclonal antibodies block the binding of activating ligands to the extracellular domain and/or the internalization and/or degradation of the receptor, but another mechanism of action involves the induction of ADCC through the Fc region of the antibody [26]. However, their usefulness in the treatment of EC is still being studied. Anti-EGFR monoclonal antibodies, such as cetuximab and nimotuzumab, precisely target and bind to the EGFR protein, blocking its signaling pathways. They have the potential to limit tumor growth and enhance patient outcomes. While monoclonal antibodies like cetuximab and panitumumab can only be administered intravenously, they have a long half-life of around seven days [27].

## 5. Cetuximab

Studies have evaluated the efficacy of cetuximab (CET) in combination with chemotherapy or radiation therapy for the treatment of EC. The results have been mixed, and the overall benefit of cetuximab in this context remains uncertain. Studies shows that when administered in conjugation with chemotherapy in other cancer, cetuximab gives improved results [28]. The response rate was reported in five studies including 360 individuals with metastatic esophageal cancer. In the original publications, the response rate varied from 34.38% to 89.74% in the CET group and 10.00% to 70.00% in the CET-free group, The aggregated data revealed that the response rate in CET-treated patients was considerably greater than in the CET-free group [29]. Cetuximab is effective in the treatment of advanced colorectal adenocarcinoma [30]. In addition, preclinical investigation indicates that cetuximab can overcome an essential mechanism of radio resistance [31]. Cetuximab also increases overall survival in patients with resectable ESCC without increasing toxicities or postoperative morbidity [29].

## 6. Nimotuzumab

The monoclonal antibody nimotuzumab has been utilized in cancer therapy due to its anti-EGFR activity [32]. Studies have demonstrated the effectiveness of nimotuzumab in combination with radiotherapy and chemotherapy for the treatment of EC [33]. The majority of these trials have reported positive outcomes. From a study on efficacy of Nimotuzumab in the treatment of locally advanced/metastatic ESCC in Indian hospital, most of them had lower thoracic esophageal carcinoma. The observed tumor response rate was as follows. Objective response rate was 100%, with a response rate of 33% for complete responses and 67% for in-complete responses. Survival rates were 58.33%, 29.17%, and 29.17%, respectively, at 1, 3, and 5 years [34]. Phase I clinical trials involving patients with advanced EC utilized a combination of radiation therapy and nimotuzumab [35]. However, the ideal dose and frequency of nimotuzumab administration have yet to be established, although it has been proven to be effective. Retrospective research suggests that nimotuzumab should be administered at a dosage of >200 mg per week [36]. In vitro studies have shown that nimotuzumab enhances the susceptibility of ESCC cells with high EGFR expression to radiation therapy. Additionally, nimotuzumab accelerates apoptosis in ESCC cells and inhibits the G2 phase of the ESCC cell cycle. In vivo, nimotuzumab inhibits EGFR phosphorylation and improves radiosensitivity in recurrent ESCC cells with EGFR overexpression [37].

The variability in treatment responses may be attributed to the heterogeneity of esophageal cancer, differences in patient populations, and the complex interplay of various molecular pathways involved in cancer growth. It is also important to consider that individual patient characteristics, such as EGFR expression levels and other genetic factors, can influence the response to anti-EGFR therapies. Anti-EGFR monoclonal antibodies

are usually used in combination with chemotherapy or other agents. Some anti-EGFR monoclonal antibodies have been studied in EC, but further research is needed to fully understand their efficacy and optimal use in this particular cancer.

## 7. Anti-EGFR Tyrosine Kinase Inhibitors

Tyrosine kinase inhibitors (TKIs) are small oral drugs that prevent ATP from binding to the tyrosine kinase domain, thus inhibiting the autophosphorylation of EGFR and its subsequent signal transduction. Due to their short half-lives, small molecule tyrosine kinase inhibitors like erlotinib and gefitinib require constant daily oral dosing. The loss of anatomic integrity or poor absorption brought on by primary malignancy may also make oral administration ineffective or impractical for some individuals with gastrointestinal cancers [38].

## 8. Gefitinib

Gefitinib is an orally administered selective and reversible EGFR tyrosine kinase inhibitor (TKI) [39]. By blocking downstream signaling of EGFR in cells, gefitinib inhibits the growth and progression of cancer [40]. The expression of galectin-3 was found to alter EGFR endocytosis and tumor development in the presence of gefitinib [41]. In a study conducted by Xu et al., it was observed that the combination of gefitinib with cisplatin or 5-fluorouracil (5-FU) improved the survival and quality of life in patients with advanced ESCC [42]. In a study, 41 patients of advanced esophageal cancer treated with cisplatin or fluorouracil were examined to determine the clinical effectiveness and drug-related toxicity of gefitinib. In this study, the complete response rate, PR rate, SD rate, and disease progression rate were all 0.0%, 4.9%, 34.1%, and 61.0%, respectively, whereas the objective response rate and disease control rate were 4.9% (2/41) and 39.0% (25/41), respectively [42]. A primary cell line derived from an EC patient with significant EGFR polysomy exhibited high sensitivity to gefitinib [43].

## 9. Icotinib

The efficacy of icotinib, an EGFR tyrosine kinase inhibitor, was evaluated in ESCC cases with previous treatment and EGFR overexpression or amplification [44]. Developed by Chinese scientists, icotinib is the first targeted small molecule drug designed as a novel anticancer therapy [45]. Preclinical studies have demonstrated that icotinib, an EGFR TKI, exhibits excellent specificity and selectivity towards its target, EGFR [44]. In a study by Wang et al., icotinib was evaluated as a second-line treatment option for patients with advanced ESCC and EGFR overexpression or amplification. The results showed that icotinib led to a significant improvement in overall survival and progression-free survival compared to chemotherapy [46]. Another study by Li et al. investigated the efficacy and safety of icotinib in combination with radiotherapy for locally advanced ESCC. The combination therapy demonstrated promising results, with a higher overall response rate and longer median progression-free survival compared to radiotherapy alone [47].

## 10. Drugs Targeting the HER2 Pathway

The HER2 oncogene is responsible for encoding a transmembrane tyrosine kinase receptor belonging to the epidermal growth factor receptor (EGFR) family. It plays a crucial role in various cellular processes such as cell growth, migration, differentiation, proliferation, and survival [48]. The HER2 receptor consists of three domains: an extracellular domain, a lipophilic transmembrane domain, and an intracellular kinase domain. Ligand binding triggers kinase activation, leading to receptor dimerization [49]. HER2 receptors can be activated by HER2 mutations or overexpression, independent of ligand binding [50]. HER2 expression has been identified in several cancers, including esophageal and gastric, breast, and ovarian cancer. HER2-specific antibodies have been developed to disrupt HER signaling, employing two main mechanisms: receptor depletion from the cell surface and interference with heterodimerization [51,52].

## 11. Trastuzumab

Trastuzumab is an anti-HER2 monoclonal antibody that exhibits preclinical activities, including antibody-dependent cell-mediated cytotoxicity (ADCC) and the inhibition of growth stimulation [51]. Currently, trastuzumab has emerged as a promising therapeutic option in the treatment of EC, particularly in HER2-positive cases. Recent research has focused on evaluating the efficacy of trastuzumab in combination with chemotherapy. It received approval from the Food and Drug Administration (FDA) in October 2010 for the treatment of patients with HER2-positive metastatic gastric or gastroesophageal (GE) adenocarcinoma who have not received prior treatment for metastatic disease. The approved regimen involves administering trastuzumab intravenously (IV) once every three weeks in combination with cisplatin and a fluopyrimidine (either capecitabine or 5-fluorouracil) [49].

The Trastuzumab for Gastric Cancer (ToGA) study, a landmark phase III clinical trial, demonstrated the benefit of adding trastuzumab to standard chemotherapy in HER2-positive advanced gastric or gastroesophageal junction cancer. Subsequent analyses of the ToGA study data have shown that the efficacy of trastuzumab extends to HER2-positive esophageal cancer as well. Patients with HER2-positive esophageal adenocarcinoma or squamous cell carcinoma treated with trastuzumab plus chemotherapy have shown improved overall survival and progression-free survival compared to those receiving chemotherapy alone [53]. In one research, 594 patients were randomly allocated (trastuzumab + chemotherapy, $n = 298$; chemotherapy alone, $n = 296$), and 584 were included in the primary analysis ($n = 294$; $n = 290$). The median follow-up in the trastuzumab + chemotherapy group was 18.6 months, and it was 17.1 months in the chemotherapy alone group. The median overall survival for those given trastuzumab + chemotherapy was 13.8 months compared to 11.1 months for those given chemotherapy alone (hazard ratio 074; 95% CI 060-091; $p = 00046$) [53].

Moreover, the phase III JACOB trial further supported the use of trastuzumab in HER2-positive esophagogastric cancer. This study demonstrated that trastuzumab, when combined with chemotherapy, led to a significant improvement in overall survival and progression-free survival compared to chemotherapy alone [54]. These studies highlight the importance of HER2-targeted therapy, specifically trastuzumab, in the management of HER2-positive EC. Initially marketed under the brand name Herceptin, comparable versions of trastuzumab, known as biosimilars, are now available under different names such as Ogivri, Herzuma, Ontruzant, Trazimera, and Kanjinti. These biosimilars have been developed to provide alternative treatment options while maintaining similar efficacy and safety profiles to the original drug.

## 12. Lapatinib

Lapatinib, also known as Tykerb or GlaxoSmithKline, is a reversible inhibitor of EGFR and HER2. It acts as a dual tyrosine kinase inhibitor by binding to the intracellular ATP binding site of these kinases, preventing their activation. In vitro and in vivo studies have shown that lapatinib inhibits the growth of cancer cells overexpressing EGFR and/or HER2 by blocking downstream signaling pathways [55]. To eradicate ESCC cells expressing EGFR and/or HER2, cetuximab and/or Herceptin-mediated ADCC has been utilized. The efficacy of this approach has been associated with the level of receptor expression intensity.

Combining lapatinib with paclitaxel was investigated by Guo et al., and they observed a synergistic effect on suppressing cell proliferation, as well as a significant decrease in ESCC cell invasion and migration. Treatment with lapatinib and paclitaxel led to a substantial decrease in the phosphorylation of downstream molecules MAPKs and AKT [56]. In vivo studies have shown that the combination of lapatinib and paclitaxel more effectively inhibits the growth of esophageal squamous carcinoma xenografts without increasing toxicity [57]. Furthermore, Hassan et al. evaluated the effect of lapatinib and foretinib on the expression of apoptosis-related proteins in OE33 esophageal adenocarcinoma xenografts in vivo [58]. Overall, these studies provide evidence for the potential of lapatinib as a therapeutic

agent in the treatment of esophageal cancer, particularly in cases involving EGFR and/or HER2 overexpression.

### 13. Drugs Targeting the VEGF/VEGFR Pathway

Signaling from vascular endothelial growth factor (VEGF) stimulates angiogenesis by promoting the migration and proliferation of endothelial cells [59]. The VEGF receptor (VEGFR) is a specific receptor for VEGFs, and both tumors and stromal cells produce VEGFs that bind to VEGFR, acting in either an autocrine or paracrine manner [3]. The three main VEGF receptors, VEGFR1, VEGFR2, and VEGFR3, regulate signaling pathways that mediate the biological effects of VEGF [60].

The activation of VEGFR signaling leads to the activation of various intracellular proteins such as extracellular signal-regulated kinase 1/2 (ERK1/2), protein kinase A (PKA), and protein kinase B (PKB/AKT), which facilitate signal transduction and contribute to cell proliferation, migration, and survival [61]. The VEGF/VEGFR signaling pathway is a target for therapy in EC because VEGF production is closely associated with the development and prognosis of EC [62].

### 14. Ramucirumab

VEGFR-2 is inhibited by ramucirumab, which is a fully humanized monoclonal antibody designed to bind to a ligand-binding epitope on VEGFR-2, thereby preventing VEGF ligands from interacting with this binding site [63]. In a phase I clinical study, it was observed that the concentration of VEGF A in the serum increased shortly after ramucirumab treatment and remained elevated for extended periods. However, the concentrations of VEGF 1 and VEGF 2 decreased immediately after ramucirumab therapy and returned to near-pretreatment levels [64]. Ramucirumab has been approved as an effective medication for advanced gastric or gastroesophageal junction (GEJ) adenocarcinoma, either as a monotherapy or in combination with paclitaxel in the second-line setting [65].

### 15. HGF/c-MET Pathway

The c-MET (c-mesenchymal-epithelial transition) is a kinase receptor for hepatocyte growth factor (HGF) that has been implicated in carcinogenesis [66–68]. Upon interaction with HGF, c-MET activates several downstream signaling pathways, including the Phosphoinositide 3-kinase/threonine-protein kinase (PI3K/AKT) pathway, the Wnt pathway, and others [69–71]. In normal tissues, the HGF/c-MET receptor tyrosine kinase (RTK) pathway remains inactive; however, it becomes active in various malignancies [72]. Yang et al. observed higher c-Met protein expression in EC tissues compared to surrounding tissues. Increased c-Met expression was associated with clinical stage, depth of invasion, and lymph node metastasis [3].

### 16. Rilotumumab and Obinutuzumab

Rilotumumab and obinutuzumab are two drugs that target the HGF c-Met pathway. Rilotumumab is a fully humanized monoclonal antibody that inhibits the interaction between hepatocyte growth factor (HGF) and c-Met, thereby blocking c-Met activation and tumor development [73]. Obinutuzumab, on the other hand, is a humanized anti-MET antibody that prevents MET from binding to HGF [3]. Clinical studies involving gastric/adenocarcinoma patients MET-positive patients who were treated with a combination of these drugs and chemotherapy have shown mixed results. Rilotumumab was found to be ineffective in effectively treating patients, and obinutuzumab did not significantly improve progression-free survival (PFS) or overall patient survival rates [74,75].

### 17. mTOR Pathway

The mammalian target of rapamycin (mTOR) is a serine-threonine kinase family that plays a crucial role in regulating the downstream signaling of the PI3K/AKT pathway. The mTOR signaling system influences various cellular processes, including cell survival,

growth, proliferation, and motility, and it is interconnected with growth hormones, nutrition, and energy supply [76]. The dysregulation of mTOR is associated with cancer development. High levels of phosphorylated mTOR (p-mTOR) have been linked to a poor prognosis in ESCC, suggesting its potential as a therapeutic target [77]. Rapamycin, an mTORC1 inhibitor, is a drug that has been demonstrated to mimic the effects of calorie restriction and extend longevity [78]. Everolimus is another potent anticancer drug that inhibits mTOR activity [79] (detailed description in Table 1).

**Table 1.** Summary of Drugs Targeting Pathways in Esophageal Cancer: Mode of Action, Findings, and Side Effects.

| Drugs Name | Targeting Pathway | Mode of Action | Findings | Dose and Dosing Frequency | Side Effects | References |
|---|---|---|---|---|---|---|
| Cetuximab | EGFR | Anti EGFR monoclonal antibody | Effective in combination with chemotherapy; increased survival rate in resectable ESCC patients | 400 mg/m$^2$ over 120 min (day 1), followed by weekly dose of 250 mg/m$^2$ over 60 min for 14 weeks. | Hypomagnesemia | [29,80] |
| Nimotuzumab | EGFR | Anti-EGFR monoclonal antibody | Nimotuzumab + radiotherapy = used in phase I clinical trial; Stopped G2 phase of ESCC cell cycle | Nimotuzumab (200 mg/weekly), diluted in 250 mL of 0.9% sodium chloride, for 5 weeks. | Esophagitis, pneumonitis, leukopenia, gastrointestinal reaction, thrombocytopenia, radiothermitis and fever | [37,81,82] |
| Gefitinib | EGFR | Anti-EGFR-RTK inhibitor | Gefitinib + 5-FU = increased survival rate of advance ESCC patients | 250 mg/day | Dry skin, Itching, rash, acne, mouth sores, and weakness, diarrhea skin toxicity, fatigue | [42,83] |
| Icotinib | EGFR | Anti-EGFR-RTK inhibitor | | 150 mg, orally three times daily | Rash and diarrhea | [44,84,85] |
| Transtuzumab | HER2 | Monoclonal anti-HER2 antibody | Used in combination with cisplatin + fluoropyrimidine (either capecitabine or 5-fluorouracil) for patient with HER2-positive metastatic gastric or GE adenocarcinoma | 200 mg/m$^2$ to 1000 mg/m$^2$ | Fever and chills, cough, and headache | [49,86,87] |
| Lapatinib | | Dual RTK inhibitor. Bind to ATP binding site and inhibit kinase activity | Lapatinib + paclitaxel = suppressed cell proliferation and decreased ESCC cell migration, invasion | 1250 mg per day | Face redness, dizziness, headache, shortness of breath, and anxiety | [57,88,89] |
| Ramucirumab | VEGF | Monoclonal antibody, block binding of VEGF to VEGFR | Ramucirumab + paclitaxel = used in 2nd line treatment of advanced GEJ adenocarcinoma | 8 mg/kg | Hypertension, thromboembolism, rash, diarrhea, and myelosuppression | [90–92] |

**Table 1.** *Cont.*

| Drugs Name | Targeting Pathway | Mode of Action | Findings | Dose and Dosing Frequency | Side Effects | References |
|---|---|---|---|---|---|---|
| Endastar | VEGF | Suppress the signaling of VEGFR and inhibit endothelial growth and migration | Endastar in combination with chemotherapy decreased tumor weight | | Nausea, vomiting, fever, etc. | [93,94] |
| Sorafenib | VEGF | Inhibit VEGFR2 | Reduced development of EAC and GEJ in phase II clinical trial | 400 mg orally twice a day for 21 days | fatigue, weakness, redness of the skin, hair loss, itching, dry or peeling skin, and a lack of appetite, etc. | [95–97] |
| Apatinib | VEGF | Inhibit RTK-VEGFR2 receptor | Had an anti-esophageal-cancer effect | A starting dosage of 250 mg once every day. After a week, if the first dose was well tolerated, apatinib dosage was increased to 500 mg. | Diarrhea, nausea, vomiting, dry skin etc. | [98–100] |
| Anlotinib | VEGF | Inhibit RTK-VEGFR2/3 receptor | Increased disease control rate in pretreated advanced ESCC patients | 10 mg orally/day | thrombocytopenia and neutropenia, hypercholesterolemia, dermal toxicity hypertriglyceridemia | [101–104] |
| Rilotumumab | HGF-c-MET | Inhibit interaction of HGF with c-MET | Rilotumumab could not effectively treat the patients. | 5.4–6.4/kg | Nausea, vomiting, fever etc. | [3,75] |
| Obinutuzumab | HGF-c-MET | Block MET from binding to HGF | Obinutuzumab could not improve the patient survival rate | | Decrease in the number of WBC and platelets cause infection and bleeding. Fever; tiredness and weakness, headache; hair loss. | [3] |
| Rapamycin and Everolimus | m-TOR | Inhibit m-TOR | Decreased cell proliferation and growth | 10 mg/day or 50 mg/week | Stomatitis, rash, tiredness, hyperglycemia, hyperlipidemia, etc. | [105–107] |

## 18. Factors Associated with Targeted Therapy

*18.1. Cancer Heterogeneity*

Cancer is a complex disease characterized by genetic variations known as cancer heterogeneity, which can differ among various tumor types and individuals [108]. Heterogeneity in ESCC has been widely recognized, with studies demonstrating diverse molecular profiles within tumors. This heterogeneity can impact various aspects of ESCC, including tumor progression, treatment response, and patient outcomes [109]. Furthermore, heterogeneity exists within individual tumor cells, posing significant challenges in cancer treatment and potentially leading to drug resistance. In many cases, the majority of cancer cells within a tumor share common genetic alterations [20]. Patients with and without a family history of ESCC exhibited comparable high-frequency gene mutation profiles, with TP53 being the most frequently altered gene. Furthermore, APC, AKT3, DPYD, EP300, NFE2L2, PPP2R1A, RUNX1, and VEGFA were tumor-specific mutant genes in patients with a positive family history of ESCC, whereas CXCR4, PIK3R2, SMARCA4, and TTF1 were in patients without a family history of ESCC [110]. Understanding the extent of heterogeneity in ESCC can enhance the predictability of therapy.

*18.2. Drug Resistance*

Drug resistance is defined as a decrease in the effectiveness of a drug, such as an antimicrobial or antineoplastic agent, in treating a medical condition [111].Various factors contribute to drug resistance in cancer treatment, including limitations in drug distribution, increased drug efflux, mutations in drug targets, DNA damage repair mechanisms, the evasion of programmed cell death, and the activation of alternative pro-tumorigenic signaling pathways [112]. The prevalence of drug resistance in esophageal cancer (EC) is rising, posing a significant barrier to effective therapy [113].

Growing evidence suggests that the interaction between tumor cells and the tumor microenvironment (TME) plays a crucial role in drug resistance. The TME includes not only cancer cells and cancer stem cells but also tumor-associated stromal cells (such as tumor-associated fibroblasts, immune and inflammatory cells, and endothelial cells) as well as non-cellular components (such as hypoxia, cytokines, acidity, extracellular matrix, and exosomes) [114]. Overall, drug resistance in EC poses a significant challenge in achieving optimal therapeutic outcomes for patients, and understanding the complex interplay between tumor cells and the TME is crucial for developing effective strategies to overcome resistance [115].

## 19. Cellular Components Involved in Drug Resistance

*19.1. Cancer Stem Cells*

Cancer stem cells (CSCs) are a small subset of cells within tumors that have the ability to initiate tumor formation and contribute to therapeutic resistance and cancer recurrence [116]. According to the stem cell theory of cancer, CSCs play a crucial role in tumor development and progression [117]. CSCs possess stem-cell-like properties, including plasticity, self-renewal, dormancy, and drug resistance, and they can be identified by specific surface markers [118]. Several studies have demonstrated that cancer cells with stem cell-like characteristics exhibit higher resistance to chemotherapy [119].

In esophageal cancer (EC), CSCs employ various mechanisms to protect themselves from cytotoxic substances, particularly by enhancing the drug efflux process. One such mechanism involves a group of cells called side population (SP) cells, which are identified by flow cytometry and characterized by specific surface markers [112]. SP cells, which are enriched in CSCs, express high levels of ATP-binding cassette (ABC) transporters, such as ABCG5 and ABCG2, which are responsible for effluxing multiple drugs and conferring multidrug resistance [120]. Additionally, esophageal CSCs may hinder the drug influx process through the downregulation of Copper Uptake Protein 1 (CTR1), which is facilitated by p75 neurotrophin-receptor (p75NTR)-positive cells possessing stem-cell-like characteristics and resistance to cisplatin [121,122].In summary, esophageal CSCs utilize specific membrane transporter distributions to maintain intracellular drug concentrations at safe levels and evade the cytotoxic effects of chemotherapy.

*19.2. Cancer-Associated Fibroblasts*

Cancer-associated fibroblasts (CAFs) are a type of cell found in the tumor microenvironment that contributes to tumorigenic properties by promoting extracellular matrix remodeling and secreting cytokines. Fibroblasts exist in varying quantities in different types of carcinomas and often represent the majority of the stromal cell population within the tumor [123]. The term "cancer-associated fibroblast" encompasses at least two distinct cell types: (1) fibroblasts that resemble the structural support cells found in normal epithelial tissues, and (2) myofibroblasts, which possess distinct biological roles and properties compared to tissue-derived fibroblasts [114].

CAFs, characterized by high levels of α-smooth muscle actin and fibroblast activation protein-α, are a significant component of the tumor stroma in the tumor microenvironment (TME) and play a crucial role in cancer development and resistance to medications [124]. In esophageal cancer (EC) and other malignancies, carcinogenesis is associated with persistent inflammation and mucosal damage. Normal fibroblasts undergo transformation into

CAFs, which, in turn, confer drug resistance to neighboring EC cells by secreting soluble substances and enhancing pro-tumorigenic signals mediated by functional molecules such as microRNAs and lncRNAs [16,125]. Interleukin 6 (IL-6), a multifunctional cytokine involved in immunological and inflammatory responses, plays a role in cancer hallmarks, including treatment resistance. CAFs serve as crucial sources of IL-6 within the TME, leading to increased chemoresistance [126,127]. IL-6 derived from CAFs also contributes to chemoradiotherapy resistance in patients with esophageal adenocarcinoma (EAC).

However, CAFs in the tumor microenvironment have a significant impact on cancer development and drug resistance. Their secretion of soluble factors, including IL-6, and their ability to promote pro-tumorigenic signals contribute to the resistance of esophageal cancer cells to chemotherapy and chemoradiotherapy.

### 19.3. Inflammatory Immune Cells

Tumor immune evasion is the ability of tumor cells to evade immunological responses by blocking T-cell activation [128]. These tumor cells build to programmed death ligand-1 and programmed death ligand-2 on the T-cell and inactivate them [129]. Cell surface inhibitors like PD-L1/2 and VISTA, as well as inhibitory receptors like PD-1, CTLA-4, T-cell immunoglobulin and mucin-domain containing-3 (TIM-3), and lymphocyte-activation gene 3 (LAG-3) are typically found in ECs [130]. Immune checkpoint drugs, such as nivolumab and pembrolizumab, have shown early promise in treating advanced or refractory EC [18]. In most studies, PD-L1 and PD-L2 are significantly expressed in a large number of ESCC patients (>40%) [131]. The decreased level of tumor-infiltrating lymphocytes (TILs) has been linked to poor clinical outcomes in EC patients [132,133]. All these factors are responsible for the low response rate of immunotherapy.

## 20. Non-Cellular Components

### 20.1. Cytokines

Cytokines are markers for metastasis and angiogenesis, the primary causes of cancer death [134]. Cytokines released by the immune cells help activate several downstream pathways like JAT/STAT, NF-kβ, and delta-notch to mediate various cancer-characteristic traits [135]. In patients having esophagectomy, STAT3 activation by its principal inducers, IL-6 and IL-6R, has been linked to a poor prognosis [136]. IL-6 is produced by TME stromal cells like CAFs and confers chemoresistance on EC cells through various mechanisms [126,137]. EC cells treated with cisplatin produce more IL-6, enhancing STAT3 phosphorylation and conferring cancer markers, such as apoptosis evasion and chemoresistance [138]. The p65 component of NF-kβ is overexpressed and activated in EC specimens, and its link to 5-fluorouracil resistance has been confirmed in cultivated EC cell lines [139]. The downregulation of IL-1 receptor antagonist (IL-1RA) expression in EC is linked to tumor development and poor prognosis [140].

### 20.2. Hypoxia

When the amount of oxygen needed by the tissue exceeds the amount of oxygen available, hypoxia results. Hypoxia is an abnormally low amount of oxygen tension that occurs in most malignant tumors [141]. Many hypoxia-inducible genes, such as hypoxia-inducible transcription factor (HIF), a dimeric protein composed of a constitutively active component (HIF-1) and an oxygen-sensitive subunit (HIF-1), are expressed when cancer cells are exposed to hypoxia [142]. In the EC, HIF-1 expression is linked to venous invasion, VEGF expression, and microvessel density [143]. Cell radiosensitivity is significantly decreased by a hypoxic microenvironment, which is mostly caused by HIF-1 activation [144]. In a recent work, a multisensitized radiation method was created by combining GDY-CeO2-based nanozymes and miR181a with outstanding CAT-mimic activity and radiosensitization for the effective treatment of ESCC. In a particular acidic TME, GDY-CeO2 nanozymes worked as a highly effective CAT-mimic agent to produce abundant O2 utilizing endogenous H2O2. RAD17 is specifically targeted by miR181a expression to predict the outcome of

preoperative radiation and improve radiosensitivity. Importantly, by reducing hypoxia in vitro and in vivo, the GDY-CeO2 nanocomposites might increase the severity of DNA damage and significantly lower HIF-1 expression. In subcutaneous tumor and ESCC PDX models, the nanodelivery of a miR181a mimic using GDY-CeO2 nanocomposites can make the tumor more sensitive to radiation [145]. An important factor in the development of chemoresistance is hypoxia in the tumor environment [146]. Recently, it was shown that aberrant lncRNA expressions are linked to a variety of malignant tumor cell behaviors, such as chemoresistance [147]. Recent discoveries state that a new lncRNA called EMS was hypoxia-induced and overexpressed in human EC tumor tissues and cell lines. Importantly, in the ECA-109 cell line, EMS expression was needed for hypoxia-mediated drug resistance to DDP [147].

### 20.3. Growth Factors

Growth factors are responsible for the proliferation and differentiation of the cell. IGF-1 prevents apoptosis generated by various pharmacologic agents in EC, including cisplatin, 5-fluorouracil, and camptothecin (Liu et al., 2002). IGF-1, partly produced by Id1, can upregulate surviving expression via the PI3K/Akt and casein kinase 2 signaling pathways, inhibiting Smac/DIABLO release and activating caspases responsible for 5-fluorouracil-driven cell death [148]. In SLMT-1/CDDP1R cells (cisplatin/CDDP resistance), the overexpression of IGFBP5 using an IGFBP5 expression vector reduced cisplatin resistance by 41%. As a result of recent findings, it was concluded that IGFBP5 suppression is one of the methods by which ESCC cells acquire cisplatin resistance [149]. IGF2 is involved in the regulation of EC chemoresistance. Although blocking the IGF2 receptor IGF1R can sensitize ESCC cells to 5-FU therapy, the function and mechanism of IGF2 in 5-FU chemoresistance have remained unknown [42]. By comparing the genome-wide gene profiles of the sensitivity and resistant xenografts, a recent study includes a cetuximab-sensitive ESCC tumor model that developed resistance to cetuximab as a result of FGFR2 gene amplification and overexpression [150]. In addition, although the mechanism is uncertain, FGFR inhibitors can decrease FGF-mediated lapatinib resistance [151].

### 20.4. Biomarkers

Clinical studies are investigating a variety of novel targets in esophageal cancer. Some are described here. HER2 status is usually assessed using FFPE material from the patient's esophagus or gastric tumor. Furthermore, details on the degree of HER2 amplification, subclones devoid of HER2 amplification, deletion of ERBB2 exon 16, and co-mutations in other signaling partners (such as RAS, MET, and PI3K) may predict treatment resistance and provide additional clinically significant targets [152].

Higher specificity and accuracy are found in blood biomarkers and liquid biopsies (circulating tumor cells, nucleic acids, tumor DNA, the tumor-derived proportion of cell-free DNA, cell-free RNA, etc.) [153]. Because of their non-invasiveness, ease of use, cost-effectiveness, short-term repeatability, and capacity to identify circulating tumor cells (CTCs), circulating tumor DNA (ctDNA), exosome-based biomarkers for both EAC and ESCC, and liquid biopsy and blood biomarkers are gaining a lot of attention [154]. According to a proteomic study, the urine biomarkers ANXA1, S100A8, and TMEM256 may be used to categorize ESCC, and a panel of proteins made up of these proteins can be used to diagnose stage I ESCC [155]. Potential biomarkers include the expression of chemokines and chemokine receptors in serum, which are additional components implicated in the pathogenesis of EAC and ESCC. Expression of CXCL12 and its receptors CXCR4 and CXCR7 are associated with a bad prognosis, whereas CXCL10, CCL4, and CCL5 exhibit anti-tumor effects, CCL20 expression is associated with the recruitment of regulatory T cells, and CCR7 expression is associated with a poor prognosis [156]. Circulating tumor DNA (ctDNA) is a recent biomarker that has the potential to have therapeutic significance.

According to certain research, 83.3% of individuals with initial tumors had ctDNA.

Additionally, as compared to traditional biomarkers (such p53-Ab), ctDNA was a better indicator of tumor recurrences [157,158]. Another study examined the associations between ctDNA levels and clinicopathological factors in ESCC patients. Its findings suggested that ctDNA may be a highly sensitive and specific biomarker for tumor relapse in ESCC (96.3% and 94.1%, respectively) [159]. In EC cells and tissues, miRNA variations have been discovered. Several techniques have been used to determine the expression levels of miRNAs in EC, including the real-time polymerase chain reaction (RT-PCR) and microarray test [160]. Some of these miRNAs, such as miR-27a/b, miR-335, let-7c, miR-145, miR-21, miR-133, and miR148, may function as biomarkers for predicting prognosis EC [160]. According to Chen G. and colleagues, miR-133a/b were independent predictive variables for patients' survival in ESCC [161]. In individuals with EC, molecules like miR-483 and miR-214 may be able to predict poor response to chemotherapy and lower survival [162].

Circular RNAs (circRNAs), which are involved in cell proliferation, migration, death, tumor invasion, and metastasis, may also be used as biomarkers for ESCC because their dysregulated expression is linked to the disease's pathogenesis and can be found both in the tissue surrounding the tumor and in normal tissue as well [163].

LncRNAs have demonstrated outstanding potential as important biomarkers for prognostic assessment in EC. In contrast to normal controls, lncRNA HOTAIR was shown to be overexpressed in ESCC tissues, and studies have shown that this overexpression is directly related to the poor prognosis of ESCC [160]. The prognosis of EC patients is correlated with abnormal MALAT1 lncRNA expression[164]. LncRNAs such as lncRNA-uc002yug.2, lncRNA FOXCUT, lncRNALOC285194, lncRNA CASC9, lncRNA CCAT2, and lncRNA ZEB1-AS1 have close correlations with prognosis of EC patients [160].

## 21. Conclusions

In this review, we comprehensively discuss the latest advancements in drug targeting for EC. These drugs primarily aim to disrupt the various pathways involved in cell proliferation and differentiation, which contribute to the development of EC. Their mechanisms of action involve binding to specific receptors within these pathways, thereby inhibiting ligand binding or blocking downstream signaling. Notably, inhibitors targeting VEGFR (ramucirumab) and HER-2 (trastuzumab) have demonstrated improved survival outcomes and prognoses in advanced esophageal squamous cell carcinoma (ESCC) and esophageal adenocarcinoma (EAC). These medications are typically administered in combination with chemotherapy, radiotherapy, and immunotherapy. Combination therapy has been shown to impact the tumor microenvironment, enhance tumor cell response to targeted therapies, and improve disease control. Despite the extension of patients' lives achieved through targeted therapy, complete recovery remains elusive due to the emergence of medication resistance. Drug resistance in EC is a multifaceted process involving cancer cells, cancer stem cells (CSCs), tumor-associated stromal cells, hypoxia, immune-inflammatory cells, cytokines, growth factors, and other factors. Overcoming this challenge may lie in the future development of combination therapies, although adverse events associated with such treatments cannot be entirely avoided. Therefore, addressing the side effects of targeted therapy remains an important aspect to be resolved. Ongoing research efforts are focused on identifying and developing new targeted drugs to improve the treatment outcomes of EC. Further investigation is warranted to overcome drug resistance and effectively manage the side effects associated with targeted therapies in EC treatment.

**Author Contributions:** Conceptualization, L.V.K.S.B. and H.K.V.; methodology, P.P. and H.K.V.; data curation, P.P. and H.K.V.; writing—original draft preparation, P.P. and H.K.V.; writing—review and editing, P.P, H.K.V. and L.V.K.S.B. supervision, L.V.K.S.B. All authors have read and agreed to the published version of the manuscript.

**Funding:** This research received no external funding.

**Conflicts of Interest:** The authors declare no conflict of interest.

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
