# Peer review of "Precision Medicine Revolutionizing Esophageal Cancer Treatment: Surmounting Hurdles and Enhancing Therapeutic Efficacy through Targeted Drug Therapies"

_onco, doi:10.3390/onco3030010_

Round 1

Reviewer 1 Report

Precision Medicine Revolutionizing Esophageal Cancer Treatment: Surmounting Hurdles and Enhancing Therapeutic Efficacy through Targeted Drug Therapies" is an overview of various therapies for the treatment of esophageal cancer. The focus is largely on receptor tyrosine kinases and their ligands.  Esophageal cancer is a serious health concern and there is a need for clinicians to understand what treatment options are available to best treat their patients as well as a need to scientists to be made aware of opportunities to develop better treatments. A comprehensive review of the current treatments, their mechanisms of actions, efficacy and therapeutic limitation would be useful to both clinicians and scientists.

The article as present has significant problems. Overall, it lacks the detail and specificity that is required to be useful to clinicians or scientists. By and large, this is a superficial overview of various treatment strategies for esophageal cancer. This is a recurrent issue throughout the article, a few examples warrant highlighting. 

1. Throughout the article they discuss numerous receptors that drive the growth and metastasis of esophageal cancer. They do not provide epidemiological data regarding the incidence of mutation or overexpression of these receptors. This is critical was considering what percentage of patients would respond to each therapy.

2. Line 100. The authors indicate that the EGFR forms heterodimers, but there is no indication with what.

3. Line 108-112 discuss the use of anti-EGFR monoclonal antibodies. There are no references within this paragraph. The information is largely self-evident: the antibodies have been widely used, the antibodies target the EGFR and blocking signaling cascades. What would be useful to a reader would be 1) the mechanism of action (how do they inhibit signaling), 2) the targeted domains of the EGFR, 3) what are biomarkers of esophageal cancer (EC) that predict a successful treatment, 4) what percentage of patients respond, and 5) what success looks like (i.e., is there an increase in the 5-year survival rate).

4. Line 127: The authors state that Nimotuzumab treatment was associated with a ‘positive outcome’. What is a positive outcome? An increase in 5-year survival? Absence of detectable cancer? The lack of specifics opens the door for misinterpretation of the article.

5. Line 129: “The trials determined the maximum tolerated dosage was safe and tolerable”.  Of course, by definition, the “maximum tolerated dosage” is “safe and tolerable”. This sentence is worthless. 

6. The repeated use of “promising” does not offer much insight into the efficacy of various drugs. Further, it is not necessary when clinical trials have been performed and the authors could indicate that “X% of patients were symptom free for X. years” or whatever additional criteria could strengthen their argument.  

7. Line 201: describes HER2-positive cancers show improves “overall survival and progression -free survival compared to those receiving chemotherapy alone”. However, there are no statistics and those could have been derived from the publication.

The authors are to be commended for the subject of the article, the clear English writing, and the use of schematics.

Author Response

Reviewer 1

  1. Throughout the article they discuss numerous receptors that drive the growth and metastasis of esophageal cancer. They do not provide epidemiological data regarding the incidence of mutation or overexpression of these receptors. This is critical was considering what percentage of patients would respond to each therapy.

Answer: Thank you very much for the suggestion we have included epidemiological data in entire manuscript.

  1. Line 100. The authors indicate that the EGFR forms heterodimers, but there is no indication with what.

Answer: Thank you very much for the suggestion we have included complete sentence manuscript.

  1. Line 108-112 discuss the use of anti-EGFR monoclonal antibodies. There are no references within this paragraph. The information is largely self-evident: the antibodies have been widely used, the antibodies target the EGFR and blocking signaling cascades. What would be useful to a reader would be 1) the mechanism of action (how do they inhibit signaling), 2) the targeted domains of the EGFR, 3) what are biomarkers of esophageal cancer (EC) that predict a successful treatment, 4) what percentage of patients respond, and 5) what success looks like (i.e., is there an increase in the 5-year survival rate).

Answer: Thank you very much for the suggestion we have included complete sentence in to the manuscript.

  1. Line 127: The authors state that Nimotuzumab treatment was associated with a ‘positive outcome’. What is a positive outcome? An increase in 5-year survival? Absence of detectable cancer? The lack of specifics opens the door for misinterpretation of the article.

Answer: Thank you very much for the suggestion we have included complete sentence in to the manuscript.

  1. Line 129: “The trials determined the maximum tolerated dosage was safe and tolerable”.  Of course, by definition, the “maximum tolerated dosage” is “safe and tolerable”. This sentence is worthless. 

Answer: Thank you very much for the suggestion we have included sentence and detailed description in to the manuscript.

  1. The repeated use of “promising” does not offer much insight into the efficacy of various drugs. Further, it is not necessary when clinical trials have been performed and the authors could indicate that “X% of patients were symptom free for X. years” or whatever additional criteria could strengthen their argument.  

Answer: Thank you very much for the suggestion we have made changes in the manuscript.

  1. Line 201: describes HER2-positive cancers show improves “overall survival and progression -free survival compared to those receiving chemotherapy alone”. However, there are no statistics and those could have been derived from the publication.

Answer: Thank you very much for the suggestion we have made changes in the manuscript in included Beng et al 2010 in the manuscript which show the comparison between these two groups.

Reviewer 2 Report

Authors have written a review article titled “Precision Medicine Revolutionizing Esophageal Cancer Treatment: Surmounting Hurdles and Enhancing Therapeutic Efficacy through Targeted Drug Therapies”. The manuscript can be considered for publication after minor revision with consideration following points.

1.      In the introduction section, add information about precision medicine.

2.      The authors have discussed various antibodies used for cancer therapy, however, their clinical status, dose and dosing frequency are missing; please add.

3.      Authors are encouraged to draw an illustrative diagram of how cellular components are involved in drug resistance.

4.      Information is inadequate in the hypoxia section of the manuscript.

5.      Drug targeting section can be further improved by using https://doi.org/10.3390/pharmaceutics15030722

Author Response

Reviewer 2

Authors have written a review article titled “Precision Medicine Revolutionizing Esophageal Cancer Treatment: Surmounting Hurdles and Enhancing Therapeutic Efficacy through Targeted Drug Therapies”. The manuscript can be considered for publication after minor revision with consideration following points.

  1. In the introduction section, add information about precision medicine.

Answer: Thank you very much for the suggestion we have included sentence for the precision medicine in the manuscript.

  1. The authors have discussed various antibodies used for cancer therapy, however, their clinical status, dose and dosing frequency are missing; please add.

Answer: Thank you very much for the suggestion we have included all information table format in the manuscript.

  1. Authors are encouraged to draw an illustrative diagram of how cellular components are involved in drug resistance.

Answer: Thank you very much for the suggestion we have included figure 1 which shows that drug and target in esophageal cancer pathway in the manuscript.

  1. Information is inadequate in the hypoxia section of the manuscript.

Answer: Thank you very much for the suggestion we have included detailed information regarding hypoxia in the manuscript.

  1. Drug targeting section can be further improved by using https://doi.org/10.3390/pharmaceutics15030722

Answer: Thank you very much for the suggestion we have included detailed information in the manuscript.

Reviewer 3 Report

The review article by Pande et al comprehensively describes targeted therapies for esophageal cancer and resistance mechanisms developed by the tumor cells during therapies. To my opinion this review will gain more scientific value by addressing in more detail suppressor networks within the tumor microenvironment (in addition to those already described by the authors namely, CAFs and cancer stem cells) induced by regulatory T cells, myeloid-derived suppressor cells, T helper cells 17, and tumor-associated neutrophils. Next, the authors should also discuss the therapeutic targeting of the suppressive tumor microenvironment in esophageal cancer via immune checkpoint inhibitors alone or combined with other therapeutic approaches such as chemo and/or radiotherapies. Finally, in their paragraph “Inflammatory Immune Cells” (lines 367-378) the authors do not clearly mention which are actually these inflammatory cells. More essential information is needed about this issue.

Some minor editing should be made.

Author Response

Reviewer 3

The review article by Pande et al comprehensively describes targeted therapies for esophageal cancer and resistance mechanisms developed by the tumor cells during therapies. To my opinion this review will gain more scientific value by addressing in more detail suppressor networks within the tumor microenvironment (in addition to those already described by the authors namely, CAFs and cancer stem cells) induced by regulatory T cells, myeloid-derived suppressor cells, T helper cells 17, and tumor-associated neutrophils. Next, the authors should also discuss the therapeutic targeting of the suppressive tumor microenvironment in esophageal cancer via immune checkpoint inhibitors alone or combined with other therapeutic approaches such as chemo and/or radiotherapies. Finally, in their paragraph “Inflammatory Immune Cells” (lines 367-378) the authors do not clearly mention which are actually these inflammatory cells.

Answer: Thank you very much for the suggestion we have made changes as per the requirement and suggestion.

Thank you very much

Sincerely

Dr Prof. LVKS Bhaskar

Round 2

Reviewer 1 Report

The revised manuscript is improved but still has many of the original concerns. The overall concern is that the information in this review article lacks the specifics and details that a reader will not get a sense of what the most commonly used therapies are, which features of esophageal cancer (EC) would make the patient respond to one therapy more than another, or how much a patient can be expect to improve following therapy. 

The list below is not comprehensive, but offers specifics regarding the criticisms.

It is confusing for the reader that EGFR inhibitors are “enabling successful treatment of EC [26]” (line 109) and then stated “…their usefulness in the treatment of EC is still being studied.” (line 118) and “They have the potential to limit tumor growth and enhance patient outcomes.” (line 120). Do EGFR inhibitors treat EC? Or are EGFR inhibitors in the exploratory stage?

Iine 60. It is not clear why the authors do not give statistics for the “modest improvements in survival rates [15]” in EC survival. Closer inspection reveals that the cited reference [15] (Development of therapeutic antibodies for the treatment of diseases. Journal of Biomedical Science, 567 2020. 27(1): p. 1.) is not an epidemiological paper and does not have data related to survival rates.

Line 173. The authors state “that the combination of gefitinib with cisplatin or 5-fluoruracil improved survival and quality of life in patients“, but there is no quantification. The cited reference provides those data and would make the information more meaningful to the reader.

Line 220. Treatment “with trastuzumab plus chemotherapy have shown improved overall survival and progression-free survival compared to those  receiving chemotherapy alone” provides no quantification. Those data are readily available in the abstract of the reference.

Again, the point of adding these statistics is to give a reader a sense of which strategies are most common and which pathways have been most successful.

In the section on heterogeneity (line 312), it would be useful to point out some of the most common mutations in ESCC and if they respond to specific therapies.

HER2 inhibitors are often the first line treatment of EC. As such it should be presented first.

It should be noted that there is no know ligand for HER2.

The formatting and spelling of the table should be checked.

Author Response

Comments and Suggestions for Authors

The revised manuscript is improved but still has many of the original concerns. The overall concern is that the information in this review article lacks the specifics and details that a reader will not get a sense of what the most commonly used therapies are, which features of esophageal cancer (EC) would make the patient respond to one therapy more than another, or how much a patient can be expect to improve following therapy.

The list below is not comprehensive, but offers specifics regarding the criticisms.

It is confusing for the reader that EGFR inhibitors are “enabling successful treatment of EC [26]” (line 109) and then stated “…their usefulness in the treatment of EC is still being studied.” (line 118) and “They have the potential to limit tumor growth and enhance patient outcomes.” (line 120). Do EGFR inhibitors treat EC? Or are EGFR inhibitors in the exploratory stage?

Answer: Thank you very much for the detailed observation, As per the suggestion we have added more points in the manuscript.

Iine 60. It is not clear why the authors do not give statistics for the “modest improvements in survival rates [15]” in EC survival. Closer inspection reveals that the cited reference [15] (Development of therapeutic antibodies for the treatment of diseases. Journal of Biomedical Science, 567 2020. 27(1): p. 1.) is not an epidemiological paper and does not have data related to survival rates.

Answer: Thank you very much for the detailed observation, As per the suggestion we have removed old reference and added new reference in manuscript.

Line 173. The authors state “that the combination of gefitinib with cisplatin or 5-fluoruracil improved survival and quality of life in patients“, but there is no quantification. The cited reference provides those data and would make the information more meaningful to the reader.

Answer:  As per suggestion we have include quantification data in the manuscript.

Line 220. Treatment “with trastuzumab plus chemotherapy have shown improved overall survival and progression-free survival compared to those  receiving chemotherapy alone” provides no quantification. Those data are readily available in the abstract of the reference.

Answer: As per suggestion we have include quantification data in the manuscript.

Again, the point of adding these statistics is to give a reader a sense of which strategies are most common and which pathways have been most successful.

Answer: As per suggestion we have added statics data which give a reader sense of which pathway have been most successful.

In the section on heterogeneity (line 312), it would be useful to point out some of the most common mutations in ESCC and if they respond to specific therapies.

Answer: As per suggestion we have add some points on mutation ESCC.

HER2 inhibitors are often the first line treatment of EC. As such it should be presented first.

Answer: Thank you very much the detailed observation HER2 inhibitor first time used to treat breast cancer patient. Later on as doctor community started to include in EC cancer.

The formatting and spelling of the table should be checked.

Answer: As per the suggestion we have checked the table.

Thank you very much

Sincerely

Dr LVKS Bhaskar